# Reconstruction of Missing Electrocardiography Signals from Photoplethysmography Data Using Deep Neural Network

**DOI:** 10.3390/bioengineering11040365

**Published:** 2024-04-11

**Authors:** Yanke Guo, Qunfeng Tang, Shiyong Li, Zhencheng Chen

**Affiliations:** 1School of Electronic Engineering and Automation, Guilin University of Electronic Technology, Guilin 541004, China; guoyk1219@foxmail.com (Y.G.); lishiyong@guet.edu.cn (S.L.); 2School of Life and Environmental Sciences, Guilin University of Electronic Technology, Guilin 541004, China; tangqunfeng@guet.edu.cn

**Keywords:** miss ECG reconstruction, electrocardiography, photoplethysmography, bidirectional long short-term memory network, UNet

## Abstract

ECG helps in diagnosing heart disease by recording heart activity. During long-term measurements, data loss occurs due to sensor detachment. Therefore, research into the reconstruction of missing ECG data is essential. However, ECG requires user participation and cannot be used for continuous heart monitoring. Continuous monitoring of PPG signals is conversely low-cost and easy to carry out. In this study, a deep neural network model is proposed for the reconstruction of missing ECG signals using PPG data. This model is an end-to-end deep learning neural network utilizing WNet architecture as a basis, on which a bidirectional long short-term memory network is added in establishing a second model. The performance of both models is verified using 146 records from the MIMIC III matched subset. Compared with the reference, the ECG reconstructed using the proposed model has a Pearson’s correlation coefficient of 0.851, root mean square error (RMSE) of 0.075, percentage root mean square difference (PRD) of 5.452, and a Fréchet distance (FD) of 0.302. The experimental results demonstrate that it is feasible to reconstruct missing ECG signals from PPG.

## 1. Introduction

An electrocardiography (ECG) signal is one of the most important bioelectrical signals that is produced as a result of the cyclic contraction and expansion of the heart muscle [1]. An ECG signal is characterized by five peaks, P, Q, R, S, and T, which reflect the electrical activity of the heart and can be measured using electrodes placed on the skin, thereby providing vital information for cardiovascular pathology [2]. Because ECG signals can directly reflect cardiac electrophysiological processes, they have become essential for cardiologists to diagnose cardiac arrhythmias and other cardiac diseases. ECG plays a vital role in detecting various cardiovascular diseases and cardiac abnormalities by classifying various virtual features. However, collecting ECG signals requires attaching electrode pads to the body surface of a patient, as well as their active participation, so patient comfort is poor. A PPG signal is a signal detected using photoelectric technology that can reflect changes in the blood volume of peripheral blood vessels caused by cardiac activity. Methods for its measurement have the advantages of portability and patient comfort [3]. During the long-term measurement of ECG signals, there are two main problems: partial signal loss due to sudden loosening of electrodes and damage due to motion artifacts and various noises. In contrast, PPG is considered unobtrusive, low-cost, and convenient for continuous monitoring. Although PPG technology has become popular in healthcare monitoring [4], ECG remains the standard and fundamental method of measurement for medical diagnosis, with abundant supporting literature and research. It is known that the peak-to-peak interval of PPG is highly correlated with the R-R interval (the time elapsed between two consecutive R peaks) of ECG, suggesting the possibility of deriving ECG signals from PPG [3]. Therefore, based on these observations, we propose exploiting this correlation to reconstruct the missing ECG signals directly from PPG measurements.

Some studies have used mathematics or deep learning techniques to reconstruct ECG signals from PPG data. Three such examples are the discrete cosine transform (DCT) [5], cross-domain joint dictionary learning (XDJDL) [6], and scattering wavelet transform (SWT) [7] models, which have been proposed for reconstructing electrocardiograms from PPG based on mathematical methods. The first two studies proposed linear regression models using the correlation between PPG and ECG. However, the correlation between ECG and PPG is not linear. The last study proposed a nonlinear model using the correlation between PPG and ECG. The basis of these studies is the beat-to-beat reconstruction of electrocardiograms from PPG. The accuracy of these methods depends on the accuracy of the R wave in ECG and contraction seam extraction algorithms in PPG, which can reduce the accuracy of ECG reconstruction. The computational parametric model [8], lightweight neural network [9], deep learning models based on encoder–decoder [10], BiLSTM [11], PPG2ECGps [12], P2E-WGAN [13], CardioGAN [14], Performer [15], transformed attentional neural network [16], and banded kernel ensemble method [17] have been proposed for reconstructing electrocardiograms from PPG based on deep learning methods. In [8], the author proposed a computational parametric model that extracts features from PPG to predict ECG parameters. Although their system estimates ECG parameters with over 90% accuracy on benchmark hospital datasets, the need for complete ECG waveform reconstruction is a barrier to the widespread adoption of their system. Two studies [9,10] took the beat-to-beat reconstruction of ECG from PPG as a basis, segmenting beats based on the signal period during preprocessing. However, cycle alignment and segmentation result in loss of temporal information, such as pulse transit time and heart rate variability, which are essential clinical factors. Some studies [11,12,13,14,15,16] used segment reconstruction of ECG signals from PPG as a basis. The models proposed in the first two studies targeted specific subjects and could not be generalized to multiple subjects, representing a limitation. In [13], the correlation coefficient between the reference and reconstructed electrocardiogram was only 0.835. In [14,15,16], the authors used other evaluation metrics to verify the model performance and did not calculate the correlation coefficient between the reference and reconstructed electrocardiograms. In [17], the authors proposed a banded kernel ensemble method to convert low-quality sources (PPG) into high-quality targets (ECG). Unlike the solutions based on neural networks, this algorithm does not impose any computational burden in the transformation task after obtaining the trained model. However, in all of these studies, reconstruction was carried out when ECG signals were not missing.

There have been some studies on the reconstruction or prediction of missing physiological signals. Two studies [18,19] involved the reconstruction or prediction of missing PPG signals. In [18], missing segments were predicted using a personalized convolutional neural network (CNN) and long short-term memory (LSTM) models using the short-term history of the same channel data. In [19], the authors proposed a method for short-history prediction of missing and highly corrupted data segments of time series PPG data based on a recurrent neural network (RNN). Three studies [20,21] focus on predicting missing ECG signals. In [20], the authors proposed an interpolation method based on parametric modeling to retrieve missing samples in ECG signals. In [21], the authors proposed the prediction of missing segments of ECG signals based on a bidirectional long short-term memory recurrent neural network (LSTM-RNN). Two studies [22,23] involved the reconstruction of missing cardiovascular (ECG and PPG) signals. In [22], A novel method for reconstructing damaged segments based on signal modeling is proposed. In [23], a model-based approach is proposed to reconstruct corrupted or missing intervals of ECG signals acquired along with PPG signals. However, these studies did not utilize the correlation between ECG and PPG to reconstruct ECG signals. This study proposes a deep learning method to reconstruct missing ECG signals from PPG measurements. In the existing PPG reconstructed ECG model, there are no missing signals in the ECG signal in the training dataset. In this study, however, the ECG signals are missing from the training set. Every recording of the ECG signal in this study was missing 1 s, 2 s, 3 s, or 4 s. This study proposes a neural network model that combines a dual-UNet structure and a bidirectional long short-term memory network. The performance of this model is validated using the MIMIC III matched subset.

## 2. Materials and Methods

This section discusses the dataset used in this study, the ECG and PPG signal preprocessing procedures, the proposed deep neural network structure, and the metrics for evaluation of model performance. Figure 1 is the flowchart for the model, with the training and verification process shown in Figure 1a and the testing process in Figure 1b.

### 2.1. Dataset

The data used to test the model in this study are from the MIMIC III matched subset [24]. The MIMIC III database contains a variety of physiological signals from intensive care unit patients, with many records in this subset. In this study, 146 recordings were utilized from various subjects, including lead II ECG and PPG signals. The sampling rate of both signals is 125 Hz. The length of each record is 5 min.

### 2.2. Preprocessing

Data preprocessing includes filtering, alignment I, normalization, segmentation, dataset splitting, and generation of missing data.

Filtering. Filtering of the ECG and PPG signals. Through multidimensional comparative analysis, Liang et al. [25] found that the fourth-order-type II Chebyshev filter showed better filtering performance and significantly improved the signal quality index. Therefore, a fourth-order Chebyshev bandpass filter is applied to the PPG signal with a passband frequency of 0.5–10 Hz. Similarly, A fourth-order Chebyshev bandpass filter was applied to the ECG signal with a passband frequency of 0.5–20 Hz. Since the bandpass range of PPG is narrower than that of ECG, the passband frequency of the ECG is selected as 0.5–20 Hz.Alignment I. Align the filtered ECG and PPG signals. Since there is a time lag (i.e., pulse arrival time) between ECG and PPG, aligning the R-wave peak in the ECG signal with the systolic peak in the PPG signal can remove the time lag. The R-wave peak in the ECG signal and the systolic peak in the PPG signal were detected using the Pan–Tompkins method [26] and the block-based method [27], respectively. The third contraction peak in the PPG signal is then aligned with the corresponding R peak in the ECG signal. After the ECG and PPG are aligned, the PPG signal needs to move forward so that the aligned ECG and PPG signals take less than 300 s.Normalization and Segmentation. After aligning the data, the PPG signal is scaled to the range [0, 1]. Due to alignment, the length of signals will be less than 300 s. To ensure that each record is the same length, we consider only the first 294 s of data and ignore any data afterward. Specifically, each record is divided into 3 s.Dataset Splitting. The first 60% of each recording was used for training, the next 20% was used for validation, and the remaining 20% was used for testing.Generation of Missing Data. To obtain missing ECG data, this study considered each record with some loss. To verify the effectiveness of the model, this study tested each record with a loss of 1 s, 2 s, 3 s, and 4 s. Figure 2 shows the 6 s segment of each ECG missing either 1 s, 2 s, 3 s, or 4 s. Figure 3 shows the 6 s segment with no missing ECG signal.

### 2.3. Model Architecture

The model structure of the proposed combination of WNet and BiLSTM is shown in Figure 4. In Figure 4, the terms ‘Conv’, ‘ConvTrans’, and ‘Upsample’ represent a one-dimensional convolution layer, a one-dimensional transposed convolution layer, and an upsampling layer, respectively. ‘Constantpad’ represents one-dimensional pads, which means padding the input tensor bounds with a constant value. ‘ReLU’ and ‘Tanh’ refer to the activation functions of the corresponding convolution layers. ‘BN’ represents a one-dimensional batch normalization layer. ‘Dropout’ represents a dropout layer. ‘BiLSTM’ represents a bidirectional long short-term memory layer. The slope of the ‘Dropout’ activation is set to 0.5.

As shown in Figure 4, the proposed WNet-BiLSTM model consists of two one-dimensional convolutional UNet encoder–decoder structures [28] and a bidirectional long short-term memory network. In the proposed WNet-BiLSTM, one-dimensional convolutional layers are followed by batch normalization [29] and ‘ReLU’ activation functions [30]. The last convolutional layer of WNet-BiLSTM is directly activated by ‘Tanh’. Research on image analysis has demonstrated the better performance of the method consisting of two U-blocks than that with one [31,32]. The WNet model does not use pooling layers in the descent block but uses one-dimensional convolutional layers. The kernel size and stride of the convolution are set to 4 and 2, respectively. A one-dimensional transposed convolution layer is used in the upsampling block. The kernel size and stride of the transposed convolutional layer are set to 4 and 2, respectively. Long short-term memory (LSTM) and bidirectional LSTM (BiLSTM) are suitable for handling time series problems. BiLSTM models take longer than LSTM models to reach equilibrium but provide better performance. The BiLSTM model can effectively solve sequential and time series problems [33,34]. Research on generating ECG signals shows that the BiLSTM model is robust in generating ECG signals [35]. In our study, WNet is first used to reconstruct missing ECG signals from PPG. Compared with the WNet-BiLSTM model in Figure 4, the WNet model structure only lacks the BiLSTM layer. Secondly, the WNet-BiLSTM model reconstructs the ECG signal from PPG. Physiological signals (ECG, PPG) have time rhythm features, and the BiLSTM layer can simultaneously extract contextual information and bidirectional time rhythm features of the signal. Since the envelope of adjacent signals mainly characterizes the rhythmic features of interbeat intervals, BiLSTM is applied between the contraction and expansion paths of a specific downsampling block to characterize the signal envelope [36]. The dropout layer is added to improve the generalization ability of the model and reduce overfitting.

### 2.4. Training Options

The WNet and WNet-BiLSTM models proposed in this study are trained using the Adam optimizer. The neural network was trained for 500 epochs using a batch size of 128 pairs of ECG and PPG fragments for all recordings. The learning rate is set to 0.001 and decays by 0.1 every 100 steps. All code was implemented in Python 3.9.16, and the neural network was implemented using Pytorch 2.0.0. Both models were trained on a server with the following configuration: CPU 11th generation Intel(R) Core(TM) i7-11700 @ 2.50 GHz and GPU NVIDIA GeForce RTX 3060 Ti. The loss function used in this study is defined as follows: (1)$$Loss=\frac{1}{l}\sum_{i=1}^{l}{\left(E\left(i\right)-E_{r}\left(i\right)\right)}^{2}$$

The *Loss* function uses the mean square error. E(i) and $E_{r}\left(i\right)$ represent the *i*th sample points of the reference and reconstructed ECG signals, respectively. The variable *l* represents the sample size of the reference ECG.

### 2.5. Stitching the Reconstructed ECG Segments and Alignment II

The neural network outputs reconstructed ECG segments that are 3 s in length, which must be spliced together to form a continuous reconstructed ECG signal. When combining two ECG segments, the second ECG segment is placed after the first. The spliced signal is used as the first segment, and subsequent signal segments are further merged as the second segment. This step is repeated until all test segments in the recording are connected. The spliced ECG signals were aligned using cross-correlation. The primary intention of such alignment is to improve the assessed similarity between the reconstructed ECG signal and the reference signal.

### 2.6. Performance Evaluation

To evaluate the performance of the proposed model on both the reference ECG signal and the reconstructed ECG signal, we use Pearson’s correlation coefficient (*r*) [37], root mean square error (RMSE), Fréchet distance (FD) [38], and percentage root mean square difference (PRD) for evaluation in the test set.

Pearson’s correlation coefficient (*r*): Pearson‘s correlation coefficient is a statistical measure used to evaluate the strength and direction of a linear relationship between two variables. The absolute value of *r* is from 0 to 1. An absolute value of the correlation coefficient close to 1 indicates a strong correlation, while an absolute value close to 0 indicates a weak correlation. The formula for calculating *r* is
(2)$$r=\frac{\sum_{i=1}^{l}\left(E\left(i\right)-\bar{E}\right)\sum_{i=1}^{l}\left(E_{r}\left(i\right)-{\bar{E}}_{r}\right)}{\sum_{i=1}^{l}{\left(E\left(i\right)-\bar{E}\right)}^{2}\sum_{i=1}^{l}{\left(E_{r}\left(i\right)-{\bar{E}}_{r}\right)}^{2}}$$In the given formula, E(i) and $E_{r}\left(i\right)$ represent the individual sample points of the reference ECG signal and the reconstructed ECG signal, respectively, with both indexed by *i*. The variable *l* represents the sample size of the reference ECG. The symbols $\bar{E}$ and ${\bar{E}}_{r}$ denote the mean value of the ECG signal and the reconstructed ECG signal, respectively.

Root mean square error (RMSE): Root mean square error (RMSE) is a metric used to quantify the difference between a measured value of an ECG signal and its corresponding reconstructed value. It evaluates the degree of deviation between predicted and actual values. The closer the value of RMSE is to zero, the smaller the deviation is between the predicted and actual values. The formula for calculating RMSE is
(3)$$\mathrm{RMSE}=\sqrt{\frac{1}{l}\sum_{i=1}^{l}{\left(E\left(i\right)-E_{r}\left(i\right)\right)}^{2}}$$

Percentage root mean square difference (PRD): Percentage root mean square difference (PRD) is used to quantify the distortion between the ECG signal measurement *E* and the reconstructed signal $E_{r}$. The calculation formula of PRD is
(4)$$\mathrm{PRD}=\sqrt{\frac{\sum_{i=1}^{N}{\left(E\left(i\right)-E_{r}\left(i\right)\right)}^{2}}{\sum_{i=1}^{N}E{\left(i\right)}^{2}}\times 100}$$

Fréchet distance (FD): Fréchet distance (FD) is a measure that evaluates signal similarity by analyzing the position and order of points on the electrocardiogram signal waveform and synthesizing them into a curve. With this distance metric, the spatial arrangement and order of the data points are considered when calculating the distance between two curves, allowing for a more accurate assessment of the similarity between two time series signals. The smaller the FD, the higher the similarity between the reference ECG signal and its reconstructed ECG signal. The formula for calculating FD is
(5)$$\mathrm{FD}=\min\left(\max_{i\in Q}\left(d\left(E\left(i\right),E_{r}\left(i\right)\right)\right)\right),Q=\left[1,m\right]$$The function d(∗) represents the Euclidean distance between two corresponding points on the reference ECG signal and the reconstructed ECG signal curve. The variable *m* represents the number of sampling points. The maximum distance under this sampling is denoted as $\max_{i\in Q}\left(d\left(E\left(i\right),E_{r}\left(i\right)\right)\right)$. The Fréchet distance is the value in the sampling method that minimizes the maximum distance.

## 3. Results

After model training, we evaluated the reconstruction performance using test data. The following results are obtained from the evaluation of the WNet and WNet-BiLSTM models on the test set. The blue line in the figure represents the PPG signal, the black line represents the reference ECG signal, and the red line represents the reconstructed ECG signal.

### 3.1. WNet Model Result

We first verify the performance of the WNet model when 1 s, 2 s, 3 s, and 4 s of ECG data are missing. Figure 5 shows the model input, which is the PPG signal. Figure 6 shows the experimental results for 1 s and 2 s of missing ECG signals. Figure 6a is a comparison of the reference and reconstructed electrocardiograms when there is 1 s of missing ECG signal. Figure 6c shows a comparison of the reference and reconstructed ECG signals aligned using cross-correlation in Figure 6a. Figure 6b corresponds to the reference and ECG signals when there is 2 s of missing ECG signal. Figure 6d compares the reference and the reconstructed ECG signals after alignment using cross-correlation, as shown in Figure 6b. When 1 s of ECG signal is missing, the *r*, RMSE, PRD, and FD values of the reconstructed and reference ECG signals are 0.923, 0.055, 3.896, and 0.153, respectively. After cross-correlation alignment, the *r*, RMSE, PRD, and FD values of the reconstructed and reference ECG signals were 0.954, 0.043, 3.034, and 0.153, respectively. When the 2 s of ECG signal is missing, the *r*, RMSE, PRD, and FD values of the reconstructed ECG and reference ECG are 0.932, 0.052, 3.731, and 0.139, respectively. After using cross-correlation alignment, the *r*, RMSE, PRD, and FD values of the reconstructed and reference ECG signals were 0.953, 0.043, 3.103, and 0.139, respectively.

Figure 7 shows the experimental results when 3 s and 4 s of ECG signals are missing. Figure 7a shows a comparison of the reference and reconstructed electrocardiograms when there are 3 s of missing ECG signal. Figure 7c is a comparison of the reference and reconstructed ECG signals aligned using cross-correlation in Figure 7a. Figure 7b corresponds to the reference and reconstructed ECG signals when there is 4 s of missing ECG signal. Figure 7d compares the reference and the reconstructed ECG signals after alignment using cross-correlation, as shown in Figure 7b. When 3 s of ECG signal is missing, the *r*, RMSE, PRD, and FD values of the reconstructed and reference ECG signals are 0.927, 0.053, 3.848, and 0.169, respectively. After using cross-correlation alignment, the *r*, RMSE, PRD, and FD values of the reconstructed and reference ECG signals were 0.955, 0.042, 3.042, and 0.169, respectively. When 4 s of ECG signal is missing, the *r*, RMSE, PRD, and FD values of the reconstructed and reference ECG signals are 0.939, 0.049, 3.561, and 0.180, respectively. After using cross-correlation alignment, the *r*, RMSE, PRD, and FD values of the reconstructed and reference ECG signals were 0.949, 0.044, 3.244, and 0.180, respectively.

From Figure 6 and Figure 7, it can be determined that when 1 s, 2 s, 3 s, and 4 s of ECG signals are missing, the *r* values of the reference and reconstructed ECG signals are 0.923, 0.932, 0.927, and 0.939, respectively. When there are 4 s of ECG signal missing, the *r* value for the reference and reconstructed ECG signals is the highest. After cross-correlation alignment, the r values of both the reference and reconstructed ECG signals increase to a certain extent when there are 1 s, 2 s, 3 s, and 4 s of missing ECG signals.

Figure 8 is a box plot of the Pearson correlation coefficient, RMSE, PRD, and FD for 1 s, 2 s, 3 s, and 4 s of missing ECG signals. Here, the red wireframe represents the experimental results when 1 s, 2 s, 3 s, and 4 s of ECG signals are missing, and the blue wireframe represents the experimental results when 1 s, 2 s, 3 s, and 4 s of ECG signals are missing after using cross-correlation. The dots and horizontal lines represent the mean and median values, respectively. Experiments I, II, III, and IV represent 1 s, 2 s, 3 s, and 4 s of missing ECG signals, respectively. As can be seen from Figure 8, the mean ranges of *r*, RMSE, PRD, and FD of the reference and reconstructed ECG signals are [0.8, 0.9], [0.05, 0.1], [4, 6], and [0.2, 0.4], respectively. Figure 8 shows a more intuitive representation of the overall distribution of the model performance indicators when missing 1 s, 2 s, 3 s, and 4 s of ECG signals.

Table 1 shows the performance results of the WNet model when 1 s, 2 s, 3 s, and 4 s of ECG signals are missing. It can be determined from Table 1 that there is little difference in the *r* value of the reference and reconstructed ECG signals when the 1 s, 2 s, 3 s, and 4 s of ECG signals are missing. After using cross-correlation alignment, the r values of both the reference and reconstructed ECG signals increased by 0.25. Thus, the WNet model can reconstruct missing ECG signals from PPG, and the model performance is improved to a certain extent following cross-correlation alignment.

### 3.2. WNet-BiLSTM Model Result

In this section, we verify the performance of the WNet-BiLSTM model when 1 s, 2 s, 3 s, and 4 s of ECG data are missing. Figure 9 is the input PPG signal. Figure 10 shows the experimental results for 1 s and 2 s of missing ECG signals. Figure 10a,c shows the reference and reconstructed ECG signals without and with cross-correlation alignment, respectively, when there is 1 s of missing ECG signal. Figure 10b,d show the reference and reconstructed ECG signals without and with cross-correlation alignment, respectively, when there are 2 s of missing ECG signal. When there is 1 s of missing ECG signal, the *r*, RMSE, PRD, and FD values of the reconstructed and reference ECG signals are 0.865, 0.074, 5.177, and 0.149, respectively. After alignment using cross-correlation, the *r*, RMSE, PRD, and FD values of the reconstructed and reference ECG signals were 0.959, 0.043, 3.021, and 0.149, respectively. When there are 2 s of missing ECG signal, the *r*, RMSE, PRD, and FD values of the reconstructed and reference ECG signals are 0.883, 0.068, 4.825, and 0.180, respectively, and after using cross-correlation alignment, the *r*, RMSE, PRD, and FD values of the reconstructed and reference ECG signals are 0.954, 0.043, 3.025, and 0.180, respectively.

Figure 11 shows the experimental results when missing 3 s and 4 s ECG signals. Figure 11a,b is the reference ECG and reconstructed ECG with missing 3 s and 4 s ECG signals. Figure 11b,d is the reference ECG and reconstructed ECG aligned using cross-correlation in Figure 11a,b. The missing 3 s ECG signal uses cross-correlation alignment, and the r of the reference ECG and reconstructed ECG increases from 0.862 to 0.953. The missing 4 s ECG signal uses cross-correlation alignment, and the r of the reference ECG and reconstructed ECG increases from 0.884 to 0.955.

It can be seen from Figure 10 and Figure 11 that when 1 s, 2 s, 3 s, and 4 s of ECG signals are missing, the *r* values of the reference and reconstructed ECG signals are 0.865, 0.883, 0.862, and 0.884, respectively. When 4 s of ECG signal are missing, the *r* value of the reference and reconstructed ECG signal is the highest. When 1 s, 2 s, 3 s, and 4 s of ECG signals are missing, the use of cross-correlation alignment improves the effect of ECG signal reconstruction.

Figure 12 is a box plot of the *r*, RMSE, PRD, and FD in the absence of 1 s, 2 s, 3 s, and 4 s of ECG signals. Here, the red wireframe represents the experimental results when 1 s, 2 s, 3 s, and 4 s of ECG signals are missing, and the blue wireframe represents the experimental results when 1 s, 2 s, 3 s, and 4 s of ECG signals are missing after using cross-correlation. The dots and horizontal lines represent the mean and median values, respectively. Experiments I, II, III, and IV represent 1 s, 2 s, 3 s, and 4 s of missing ECG signals, respectively. Figure 12 shows a more intuitive representation of the overall distribution of model performance indicators when 1 s, 2 s, 3 s, and 4 s of ECG signals are missing.

Table 2 shows the performance results of the WNet-BiLSTM model when 1 s, 2 s, 3 s, and 4 s of ECG signal are missing. As can be seen from Table 2, when 1 s, 2 s, 3 s, and 4 s of ECG signals are missing, the r values of the reference and reconstructed ECG signals change. Using cross-correlation alignment, the r values of both the reference and reconstructed ECG signals improved. Thus, the WNet-BiLSTM model can reconstruct missing ECG signals from PPG, with better model performance after cross-correlation alignment.

## 4. Discussion

As far as we know, only several articles have investigated missing ECG signals [20,21,22,23]. A study proposes an interpolation method based on parametric modeling to recover lost segments of ECG signals [20]. A study proposed to predict missing segments of ECG signals based on bidirectional long short-term memory recurrent neural networks [21]. A study proposes a method to reconstruct damaged segments based on signal modeling [22]. The signals reconstructed here are physiological signals (ECG and PPG). However, they all only use ECG to predict or recover missing ECG segments. One study proposed a joint model of ECG and PPG to reconstruct ECG signals [23]. In this study, the inputs to the model were ECG and PPG signals, and the output was the ECG signal. However, this study only inputs the PPG signal into the model. As far as we know, there is currently no method to reconstruct missing ECG signals from PPG.

In this study, the proposed WNet and WNet-BiLSTM models are used to reconstruct missing ECG signals from PPG. Both models perform well in reconstructing the missing ECG signals. It can be seen from Table 1 that the WNet model has the best reconstruction effect when 1 s of ECG signal is missing. Specifically, when the 1 s ECG signal is missing in the reconstruction, the Pearson’s correlation coefficient (*r*), root mean square error (RMSE), percentage root mean square difference (PRD), and Fréchet distance (FD) of the reference ECG and reconstructed ECG are 0.825, 0.081, 5.865, and 0.297, respectively. The Pearson’s correlation coefficient (*r*), root mean square error (RMSE), percentage root mean square difference (PRD), and Fréchet distance (FD) of the reference ECG and reconstructed ECG after using cross-correlation are 0.851, 0.075, 5.452, and 0.302, respectively. It can be seen from Table 2 that the WNet-BiLSTM model has the best reconstruction effect when 2 s of ECG signal are missing. Specifically, when the 1 s ECG signal is missing in the reconstruction, the Pearson’s correlation coefficient (*r*), root mean square error (RMSE), percentage root mean square difference (PRD), and Fréchet distance (FD) of the reference ECG and reconstructed ECG are 0.820, 0.083, 5.976, and 0.288, respectively. The Pearson’s correlation coefficient (*r*), root mean square error (RMSE), percentage root mean square difference (PRD), and Fréchet distance (FD) of the reference ECG and reconstructed ECG after using cross-correlation are 0.846, 0.077, 5.554, and 0.289, respectively.

A comparison of the performance of the two models in Table 1 and Table 2 demonstrates that the WNet performs better than the WNet-BiLSTM model. Adding a BiLSTM layer did not improve the model performance. Using cross-correlation alignment on the ECG signals input to the model significantly improves the model performance.

In [23], a comparison of the reconstructed ECG signal and the reference ECG signal and a box plot of the absolute error are given. It is aimed at a single signal model and only gives a box plot of the absolute error without providing a specific value. This study focuses on the group model, and four evaluation indicators are presented to verify the model’s performance. In [23], the inputs are ECG and PPG. In this study, ECG can be reconstructed by inputting only PPG. Therefore, the two studies cannot be compared. However, this study has a broader applicable scope than [23].

While the two models demonstrated enhanced reconstruction of missing ECG signals, they still have limitations, as discussed below.

The correlation coefficient for the ECG signal of the reference and that reconstructed using the model proposed in this study is only 0.851. In subsequent studies, the model will be improved to obtain better model performance.The data used in this study are considered to have missing signals, but the actual missing signals may be more complex. Thus, the application of the model has certain limitations. In subsequent studies, real missing ECG signals should be used for reconstruction.Previous work has shown that the QRS complex is more important than the P and T peaks [35]. The amplitude of the R peak in the reconstructed ECG is often smaller than the true value. The loss function in this study cannot reconstruct the ECG more accurately when using only the mean square error. Therefore, in subsequent research, QRSloss can be introduced into the loss function to verify the performance of the model.Figure 13 is an arrhythmia signal selected from the data. The current dataset contains arrhythmias and normal arrhythmias. In the study, the dataset was not divided into normal and arrhythmias. In future research, we will study the correlation mechanism between photoplethysmography and electrocardiogram signals under arrhythmias and explore the correlation between the periodic and morphological changes in electrocardiogram and photoplethysmography under different types of arrhythmias.

## 5. Conclusions

In this study, a WNet model consisting of two U-shaped structures is first proposed. Then, a bidirectional long short-term memory network is added to the WNet model in establishing a second model. These two models are used to reconstruct missing ECG signals from PPG. The input for our proposed model is the PPG signal of the 3 s segment, and the output is the ECG signal of the 3 s segment. Cross-correlation alignment is used after the model splices the output ECG signals. In order to verify the performance of the model, this study compared the performance for 1 s, 2 s, 3 s, and 4 s of missing ECG signals. At the same time, the model performance before and after using cross-correlation alignment was compared. The experimental results show that it is possible to reconstruct the missing ECG signals from PPG.

## Figures and Tables

**Figure 1 bioengineering-11-00365-f001:**
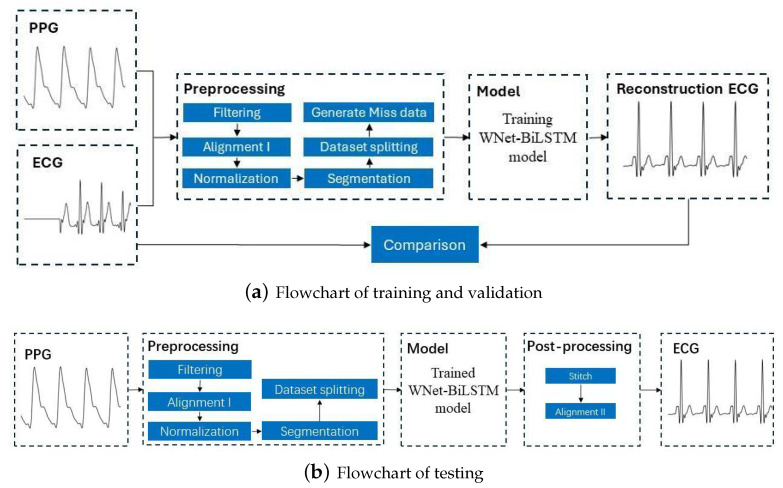
Flowchart for reconstructing missing ECG signals from PPG signals. ECG and PPG signals were segmented into segments with 375 samples. The output of the learning model is a segment of 375 ECG samples. After stitching the ECG segments, a complete ECG signal is obtained.

**Figure 2 bioengineering-11-00365-f002:**
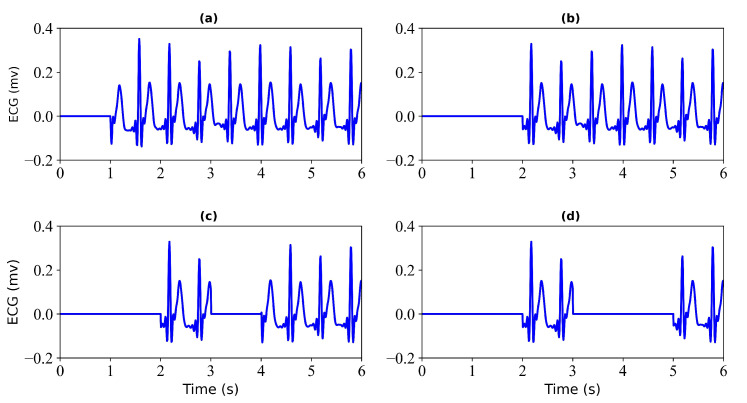
Missing ECG signal. (**a**) miss 1 s of ECG signal; (**b**) miss 2 s of ECG signal; (**c**) miss 3 s of ECG signal; (**d**) miss 4 s of ECG signal.

**Figure 3 bioengineering-11-00365-f003:**
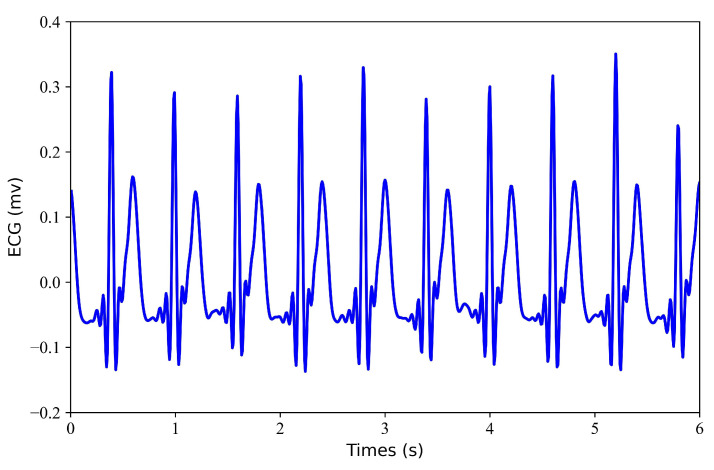
No missing ECG signal. A 6 s segment of the ECG signal is shown.

**Figure 4 bioengineering-11-00365-f004:**
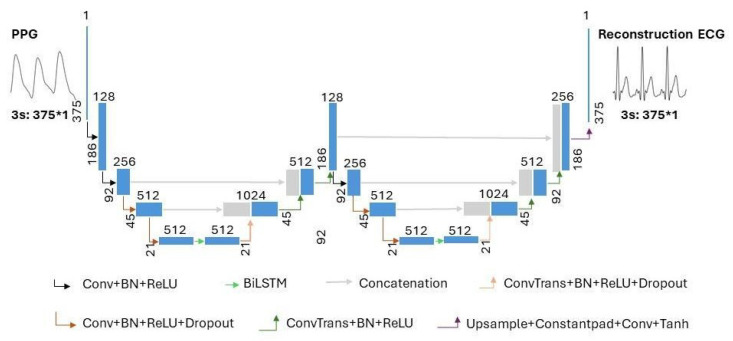
The architecture of the proposed WNet-BiLSTM model. ‘Conv’, ‘ConvTrans’, and ‘Upsample’ represent a one-dimensional convolution layer, a one-dimensional transposed convolution layer, and an upsampling layer, respectively. ‘Constantpad’ represents one-dimensional pads. ‘ReLU’ and ‘Tanh’ refer to the activation functions used in the corresponding convolution layers. ‘BN’ represents a one-dimensional batch normalization layer. ‘Dropout’ represents a dropout layer. ‘BiLSTM’ represents bidirectional long short-term memory. ’∗’ means multiplication.

**Figure 5 bioengineering-11-00365-f005:**
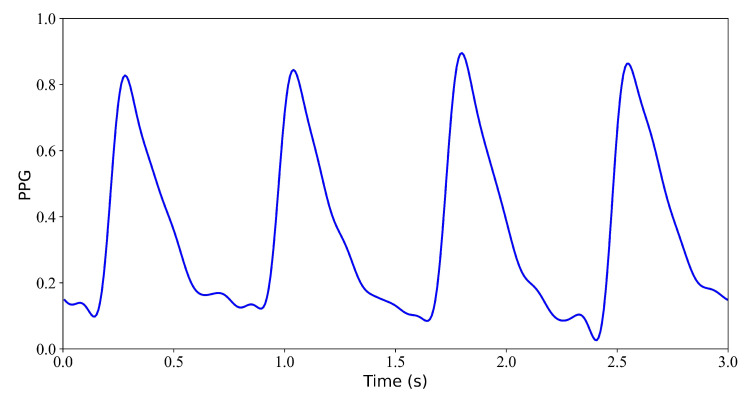
Model input: PPG signal.

**Figure 6 bioengineering-11-00365-f006:**
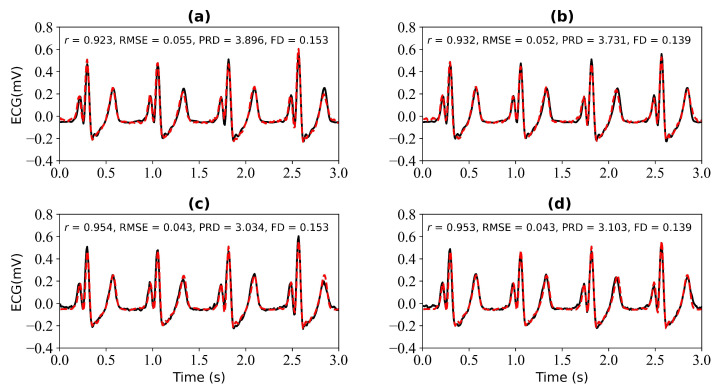
Reconstruction results for 1 s and 2 s of missing ECG signals: (**a**,**b**) represent the reconstructed missing ECG results at 1 s and 2 s, respectively; (**c**,**d**) represent the experimental results of (**a**,**b**) using cross-correlation alignment, respectively; *r*, RMSE, PRD, and FD represent Pearson’s correlation coefficient, root mean square error, percentage root mean square difference, and Fréchet distance. The black line represents the actual ECG signal (the reference ECG signal). The red line represents the reconstructed ECG signal.

**Figure 7 bioengineering-11-00365-f007:**
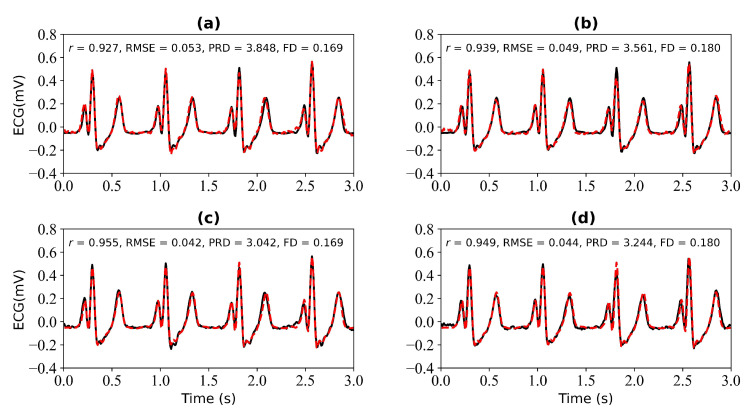
Reconstruction results for 3 s and 4 s of missing ECG signals: (**a**,**b**) represent the reconstructed missing ECG results at 3 s and 4 s, respectively; (**c**,**d**) represent the experimental results of (**a**,**b**) using cross-correlation alignment, respectively; *r*, RMSE, PRD, and FD represent Pearson’s correlation coefficient, root mean square error, percentage root mean square difference, and Fréchet distance. The black line represents the actual ECG signal (the reference ECG signal). The red line represents the reconstructed ECG signal.

**Figure 8 bioengineering-11-00365-f008:**
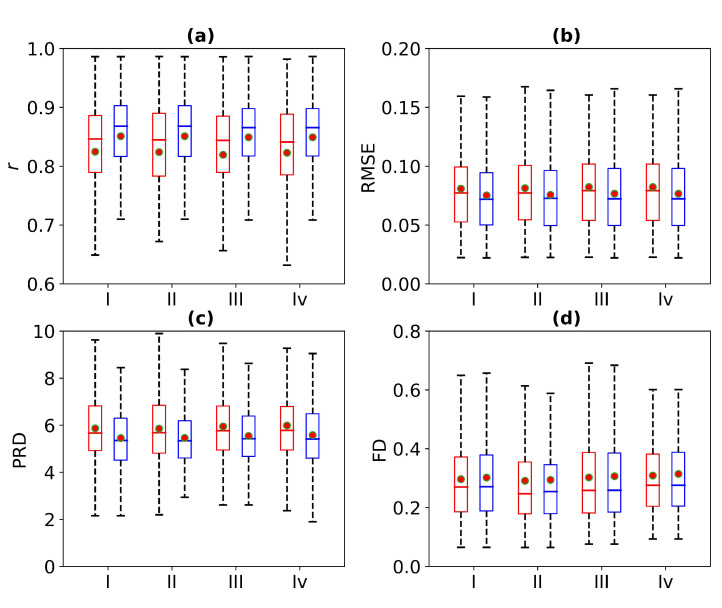
Comparison of ECG reconstruction performance for Experiments I, II, III, and IV. The statistics of (**a**) Pearson’s correlation coefficient *r*, (**b**) root mean square error (RMSE), (**c**) percentage root mean square difference (PRD), and (**d**) Fréchet distance (FD) are summarized in the box plots. The red wireframe represents the experimental results when 1 s, 2 s, 3 s, and 4 s of ECG signals are missing, and the blue wireframe represents the experimental results when 1 s, 2 s, 3 s, and 4 s of ECG signals are missing after using cross-correlation.

**Figure 9 bioengineering-11-00365-f009:**
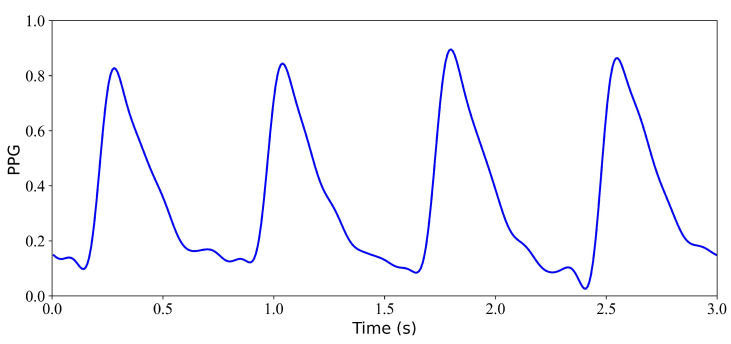
Input PPG signal.

**Figure 10 bioengineering-11-00365-f010:**
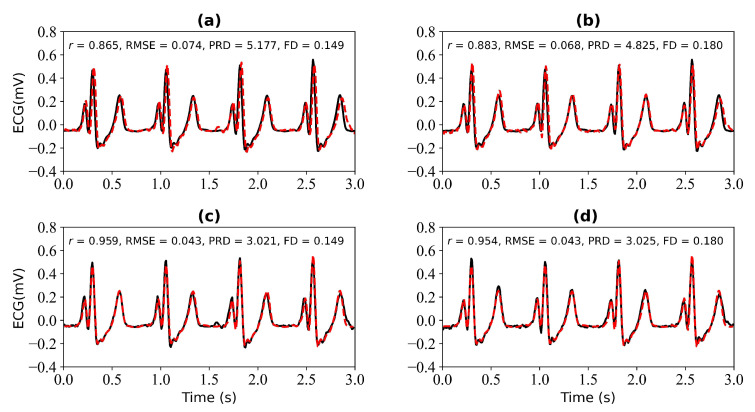
Reconstruction results for 1 s and 2 s of missing ECG signals: (**a**,**b**) represent the reconstructed missing ECG results at 1 s and 2 s, respectively; (**c**,**d**) represent the experimental results of (**a**,**b**) using cross-correlation alignment, respectively. *r*, RMSE, PRD, and FD represent Pearson’s correlation coefficient, root mean square error, percentage root mean square difference, and Fréchet distance. The black line represents the actual ECG signal (the reference ECG signal). The red line represents the reconstructed ECG signal.

**Figure 11 bioengineering-11-00365-f011:**
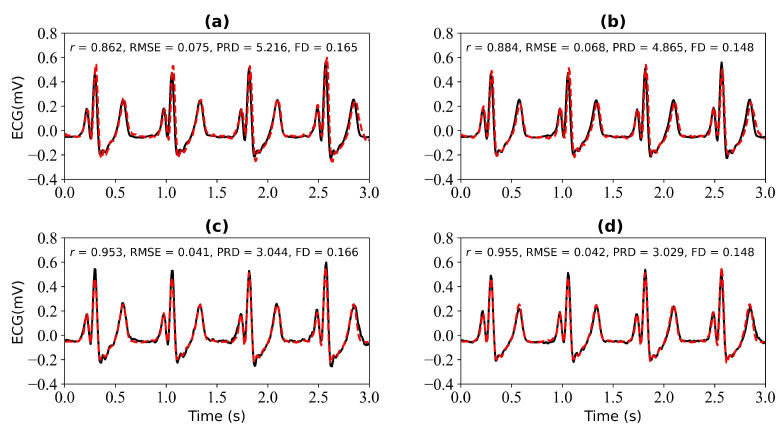
Reconstruction results for 3 s and 4 s of missing ECG signals: (**a**,**b**) represent the reconstructed missing ECG results at 3 s and 4 s, respectively; (**c**,**d**) represent the experimental results of (**a**,**b**) using cross-correlation alignment, respectively. *r*, RMSE, PRD, and FD represent Pearson’s correlation coefficient, root mean square error, percentage root mean square difference, and Fréchet distance. The black line represents the actual ECG signal (the reference ECG signal). The red line represents the reconstructed ECG signal.

**Figure 12 bioengineering-11-00365-f012:**
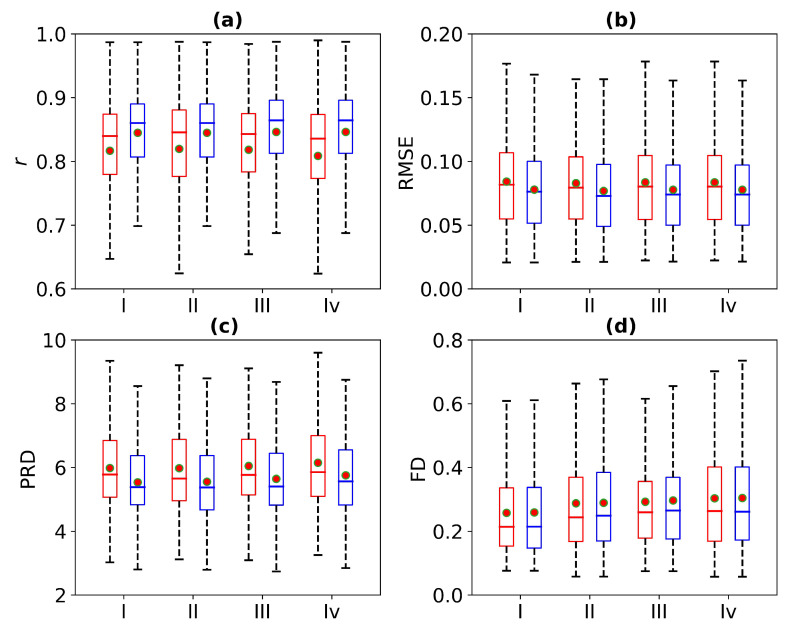
Comparison of ECG reconstruction performance for Experiments I, II, III, and IV. The statistics of (**a**) Pearson’s correlation coefficient *r*, (**b**) root mean square error (RMSE), (**c**) percentage root mean square difference (PRD), and (**d**) Fréchet distance (FD) are summarized in the box plots. The red wireframe represents the experimental results when 1 s, 2 s, 3 s, and 4 s of ECG signals are missing, and the blue wireframe represents the experimental results when 1 s, 2 s, 3 s, and 4 s of ECG signals are missing after using cross-correlation.

**Figure 13 bioengineering-11-00365-f013:**
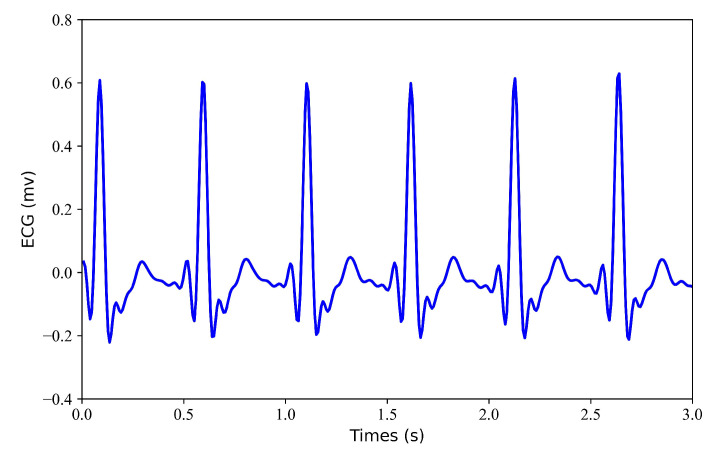
Arrhythmia ECG signal.

**Table 1 bioengineering-11-00365-t001:** Comparison of the WNet model performance when 1 s, 2 s, 3 s, and 4 s of ECG signal are missing. Note: NR stands for not reported. *r*, RMSE, FD, and PRD represent Pearson’s correlation coefficient, root mean square error, Fréchet distance, and percentage root mean square difference, respectively.

	Missing Data Length	Alignment	*r*	RMSE	PRD	FD
Experiment I	1 s	No	0.825 ± 0.110	0.081 ± 0.038	5.865 ± 1.674	0.297 ± 0.154
		Yes	0.851 ± 0.077	0.075 ± 0.034	5.452 ± 1.403	0.302 ± 0.157
Experiment II	2 s	No	0.824 ± 0.113	0.081 ± 0.038	5.857 ± 1.673	0.291 ± 0.161
		Yes	0.849 ± 0.085	0.076 ± 0.034	5.457 ± 1.418	0.294 ± 0.164
Experiment III	3 s	No	0.820 ± 0.112	0.082 ± 0.038	5.948 ± 1.697	0.303 ± 0.162
		Yes	0.845 ± 0.083	0.077 ± 0.034	5.545 ± 1.447	0.307 ± 0.165
Experiment IV	4 s	No	0.823 ± 0.106	0.081 ± 0.038	5.985 ± 1.724	0.309 ± 0.145
		Yes	0.848 ± 0.080	0.076 ± 0.034	5.581 ± 1.508	0.314 ± 0.151

**Table 2 bioengineering-11-00365-t002:** Comparison of the UNet-BiLSTM model performance, with and without alignment of the reconstructed with the reference ECG signal, and with and without alignment of the ECG signal with PPG. Note: NR stands for not reported. *r*, RMSE, FD, and PRD represent Pearson’s correlation coefficient, root mean square error, Fréchet distance, and percentage root mean square difference, respectively.

	Missing Data Length	Alignment	*r*	RMSE	PRD	FD
Experiment I	1 s	No	0.817 ± 0.102	0.084 ± 0.038	5.984 ± 1.543	0.258 ± 0.149
		Yes	0.845 ± 0.076	0.0758± 0.034	5.535 ± 1.377	0.259 ± 0.150
Experiment II	2 s	No	0.820 ± 0.107	0.083 ± 0.039	5.976 ± 1.646	0.288 ± 0.168
		Yes	0.846 ± 0.079	0.077 ± 0.034	5.554 ± 1.436	0.289 ± 0.165
Experiment III	3 s	No	0.818 ± 0.102	0.084 ± 0.039	6.048 ± 1.724	0.292 ± 0.155
		Yes	0.843 ± 0.081	0.078 ± 0.035	5.642 ± 1.624	0.297 ± 0.159
Experiment IV	4 s	No	0.809 ± 0.107	0.085 ± 0.039	6.149 ± 1.813	0.304 ± 0.171
		Yes	0.833 ± 0.089	0.080 ± 0.036	5.755 ± 1.721	0.304 ± 0.173

## Data Availability

https://archive.physionet.org/cgi-bin/atm/ATM, accessed on 22 October 2019.

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
