# Peer review of "Reconstruction of Missing Electrocardiography Signals from Photoplethysmography Data Using Deep Neural Network"

_bioengineering, 2024, doi:10.3390/bioengineering11040365_

Round 1
Reviewer 1 Report
Comments and Suggestions for Authors
In this article missing ECG signals were reconstructed using PPG. Results show some good results. Introduction is well written and understandable for the person not very sure about this field. However there are few suggestions to improve the manuscript.
1. The concept of alignment should be made very clear, especially how alignment makes signal less than 300s.
2. In Figure 1 Caption, please explain as (a) and (b), and there is some information in the caption that should be in text.
3. In Section 2.2, why a fourth order Chebyshev bandpass filter was used, any thing above why not second order. How would you defend the fact that it is not over fit for data?
4. In Figure 2, Can you please show the signal with no missing second?
5. The signal shown in Figure 5 is almost a repeating signal i.e. It is very obvious that what should be the missing signal looks like. Can't it be reconstructed using the information of repeating data?
6. The first statement, " As far as I know" in Discussion, who is "I", is He/she a first author? corresponding author. Isn't it a multiple authors manuscript?
7. Why the proposed methods not compared with existing methods?
Comments on the Quality of English LanguageReasonable
Author Response
Thank you very much for taking the time to review this manuscript. Please find the detailed responses the word file and the corresponding revisions/corrections highlighted/in track changes in the re-submitted files.

Reviewer 2 Report
Comments and Suggestions for Authors
The article requires highlighting two approaches:
- to validate the complementation of the existing ECG signal (after its deliberate removal) and to check to what extent the reconstructed waveform is consistent with the original waveform,
- evaluation of the complementation of an incomplete ECG signal (actually damaged waveforms)
in two groups of patients: conventionally healthy patients (i.e. with normal ECG waveforms) and patients with ECG deficits.
The ECG signal is a stereotyped signal, but it is crucial to reflect any (even minor) pathological changes that may be relevant for early, accurate diagnosis. For the aforementioned reasons, the reviewed study is valuable, but should be described in more detail from this angle, perhaps in a subsequent publication. On the other hand, it additionally has value within clinical simulation centres, enabling the production of findings with different pathological features that challenge pathology detection devices and training medical students and professionals. It is also crucial to consider clinical guidelines and standards (in the EU: MDR and ISO13485) in order to properly develop and implement the proposed solution.
From this point of view, three issues received too little attention in the Discussion:
- whether the restoration accuracy achieved is clinically sufficient,
- what are the current limitations of using the proposed solution,
- what are the key directions for its development/improvement.
A full answer would have raised the scientific and clinical relevance of the article.
Author Response

(The authors gave the same response as above.)

Round 2
Reviewer 1 Report
Comments and Suggestions for Authors
The authors have responded to most of my comments.
However, I have seen very minor punctuation or wording issues. For example,
In section 2.2. Filtering section Line 5, " Similarly, A fourth-oredr Chebyshev bandpass filter was applied....." The word "Similarly" should be replaced from the words like "Therefore" or "Hence" etc. The reason is before this statement authors have given a reasoning (in red) and based on this reasoning they are telling the reader that why they used 4th order filter. Word "Similarly" is not justifying it.
Similarly, the authors in again in Section 2.2 have placed multiple full stops "." at the end of few sentences.
I will suggest the authors to go through the whole manuscript once and see if they can find any minor issues related to punctuations or wording. Overall the manuscript is well written.
Satisfactory
Author Response
Thank you very much for taking the time to review this manuscript. Please find the detailed responses below and the corresponding revisions/corrections highlighted/in track changes in the re-submitted files.
